# Effects of Combined Application of Biogas Slurry and Chemical Fertilizers on Silage Corn, Soil Nutrients, and Microorganisms

**DOI:** 10.3390/microorganisms13010002

**Published:** 2024-12-24

**Authors:** Wencong Yang, Yijing Cheng, Xia Wu, Jia Zhou, Xiuping Liu

**Affiliations:** 1School of Geographical Sciences, Harbin Normal University, Harbin 150025, China; ywc@stu.hrbnu.edu.cn (W.Y.); 18944630912@163.com (Y.C.); wx98926417@163.com (X.W.); 2Center for Agricultural Resources Research, Institute of Genetics and Developmental Biology, Chinese Academy of Sciences, Shijiazhuang 050021, China

**Keywords:** biogas slurry, growth cycle, soil nutrients, soil microorganisms

## Abstract

In the greenhouse of the Chinese Academy of Sciences located on Huaizhong Road in Shijiazhuang City, Hebei Province, five fertilization treatment levels were established. These consisted of no fertilization (CK), conventional chemical fertilizer (with 100% chemical fertilizer application), and biogas slurry substitution treatments for chemical fertilizers (replacing 30%, 60%, and 100% of the chemical fertilizer nitrogen with biogas slurry nitrogen). Soil nutrient determination methods and high-throughput sequencing were employed to elucidate the correlative relationship between soil nutrients and microbial community metabolism. The results showed that in contrast to the treatment of solely applying chemical fertilizers, the combined application of biogas slurry could effectively enhance the soil nutrient content during different growth stages and promote the growth of corn plants. Among them, the replacement of 30% of the chemical fertilizer with biogas slurry exhibited the most optimal effect on augmenting soil nutrients and plant nutrient absorption. After the combined application of biogas slurry, the richness of both bacterial and fungal communities was substantially increased, and the diversity of the bacterial flora was also notably enhanced. However, the application of only biogas slurry led to a reduction in the species diversity of soil fungi. Pearson correlation analysis indicated that the Chao1 index of bacterial and fungal communities was significantly positively correlated with soil available phosphorus, available potassium, and nitrogen in the form of nitrate and significantly negatively correlated with pH. In summary, the combined application of chemical fertilizer and 30% biogas slurry was most conducive to enhancing the metabolic activity of soil microorganisms and the functional diversity of soil microbial communities, and when a relatively large amount of biogas slurry was used, it could completely substitute the chemical fertilizer.

## 1. Introduction

Over recent years, the swift expansion of large-scale livestock and poultry farming in China has led to the generation of copious amounts of livestock and poultry manure. Notably, the production of biogas slurry, a byproduct of the fermentation process of such manure, has witnessed a remarkable surge, with an annual output surpassing 1.6 billion tons [1]. The proper management and disposal of livestock and poultry manure has emerged as one of the pivotal determinants for the sustainable progression of the livestock and poultry farming sector [2]. Biogas slurry represents the liquid residue obtained following the anaerobic fermentation of diverse organic materials, including livestock and poultry manure and crop straws. As a superior organic liquid fertilizer, it harbors a rich assemblage of macronutrients such as nitrogen, phosphorus, and potassium, along with an array of micronutrients like calcium, iron, zinc, and copper. It also encompasses microbial metabolites like amino acids, vitamins, active enzymes, and hormones. Moreover, it contains substantial remnants of incompletely decomposed raw materials and microbial flora and is thereby capable of fulfilling a wide spectrum of fertilizer requirements during the growth and development of crops [3].

The substitution of biogas slurry in lieu of chemical fertilizers in agricultural production regimens manifests a multiplicity of advantages. From an economic cost perspective, biogas slurry, being derived from the anaerobic fermentation of organic substances within or adjacent to farms, can be readily procured, obviating the need for costly long-distance transportation and large-scale storage of chemical fertilizers and diminishing the reliance on expensive commercial chemical fertilizers. This, in turn, effectively curtails the direct economic outlays in agricultural production. However, an excessive combination of biogas slurry and chemical fertilizers can precipitate soil salt accumulation, structural compaction, and nutrient disequilibrium. It can also exacerbate the risk of eutrophication in water bodies, augment the residue levels of harmful substances in agricultural products, and attenuate the stress tolerance of crops, thereby exerting a host of adverse impacts on soil quality, water resources, agricultural product integrity, and crop growth [4]. Consequently, the determination of appropriate application rates and safety thresholds for biogas slurry and chemical fertilizers remains an outstanding challenge that demands urgent attention. Research has demonstrated that a judicious combination of biogas slurry and chemical fertilizers can substantially enhance soil organic carbon levels, optimize soil physical attributes, augment soil nutrient reservoirs, boost soil microbial vitality, and effectively stimulate biomass accretion in crops across different growth stages. It can also fortify the uptake and accumulation efficiencies of nutrients such as nitrogen and phosphorus by plants, thereby augmenting fertilizer agronomic efficiencies and crop yields [5,6]. Hence, the elucidation of the optimal combination ratio of biogas slurry and chemical fertilizers persists as a prominent focus of contemporary research.

Soil microorganisms constitute the linchpin of the soil ecosystem. They orchestrate the decomposition and transformation of organic matter, liberating plant-accessible nutrients and underpinning the maintenance of soil fertility. They are intricately involved in the cycling of carbon, nitrogen, phosphorus, and other elements, ensuring the seamless flow of substances within the ecosystem. As a vital component of soil activity, microorganisms play a cardinal role in soil formation, bioremediation, and the preservation of ecological equilibrium. They form the bedrock for the continuous supply of soil nutrients and exert profound influences on the structural and functional attributes of the soil ecosystem as well as the above-ground ecological processes [7,8,9]. Therefore, investigations into the ramifications of varying fertilization ratios on microbial communities hold substantial significance. A study by Yu et al. [10] divulged that biogas slurry can augment the abundance and diversity of fungal communities in the rhizosphere soil of crops, whereas its impact on bacterial communities was less pronounced. Hao Yan et al. [11] ascertained that disparate amounts of biogas slurry give rise to differential patterns of carbon source utilization by soil microbial communities. The prolonged application of biogas slurry intensifies the inter-treatment variability and augments the heterogeneity of rhizosphere microbial communities, attributable to the alterations in soil organic matter induced by the nutrient constituents of biogas slurry and the consequent modifications in carbon sources, culminating in microbial functional diversity. Galvez et al. [12] reported in their study that an appropriate dosage of biogas slurry can efficaciously ameliorate soil quality and foster a more balanced distribution of diverse nutrient elements within the soil matrix.

In the present experiment, a pot trial was conducted wherein biogas slurry was employed as a fertilizer in conjunction with chemical fertilizers. The temporal dynamics of soil nutrients and the response characteristics of soil microbial communities under diverse combination ratio treatments were meticulously dissected. A battery of multivariate statistical analysis techniques was deployed to ascertain the optimal combination ratio. The anticipated outcomes of this research are poised to furnish critical theoretical underpinnings for the innocuous treatment and resourceful utilization of livestock and poultry manure, the large-scale dissemination of biogas fertilizers in agricultural transition zones, and the optimization of the agricultural circular system revolving around biogas resources. It also holds the potential to address the vexing scientific conundrums of low soil fertility and suboptimal crop yields in agricultural–pastoral transition regions.

## 2. Materials and Methods

### 2.1. Experimental Materials

This study endeavored to simulate the actual corn cultivation scenarios in the Saibei region through pot experiments. The silage corn variety Demaiya 1, featuring a growth cycle of merely 100 days, was selected. This short growth cycle aligns well with the requirements of silage corn as livestock feed, thus prompting the comprehensive exploration of the entire growth cycle of this variety. The Saibei region, located in the northern part of Hebei Province, represents a typical agro-pastoral ecotone, integrating crop farming and livestock breeding agricultural production modalities. With the remarkable augmentation of the output value of agriculture and animal husbandry, prominent issues such as environmental resource depletion and soil fertility decline have emerged, gradually transforming this area into an ecologically fragile zone. In light of this, an experimental paradigm integrating silage corn and the livestock breeding system was meticulously devised. Silage corn serves as fodder, while livestock manure acts as the raw material for biogas slurry, thereby vigorously propelling the research progress in the agricultural application of biogas slurry. This lays a solid foundation for the rational application and scientific management of biogas slurry, effectively resolving the intractable problems of insufficient soil fertility and suboptimal crop yield increase in the agro-pastoral ecotone. The experimental soil was procured from the natural soil of the Saibei Ecological Experimental Station and was classified as a typical steppe chestnut soil (Calcic Kastanozem). Its fundamental physicochemical properties were as follows: a pH value of 8, an available nitrogen content of 150 mg·kg^−1^ (the sum of NH_4_^+^ and NO_3_^−^-N), an available phosphorus content of 15.48 mg·kg^−1^, an available potassium content of 199.5 mg·kg^−1^, a total nitrogen content of 1.85 g·kg^−1^, and an organic matter content of 27.5 g·kg^−1^. The tested biogas slurry was collected from the “Modern Animal Husbandry” station in the Saibei Management District, ensuring a close association between the source of the biogas slurry and the actual production in the research area. The physical and chemical properties of biogas slurry are shown in Table 1.This precisely reflected the impact of biogas slurry on corn cultivation and soil environment under the local agricultural production conditions. Concurrently, urea (containing 46% nitrogen), a phosphate fertilizer containing 10% phosphorus pentoxide, and potassium chloride (containing 60% potassium oxide) were respectively designated as nitrogen, phosphorus, and potassium fertilizers, furnishing standardized fertilizer conditions for the experiment. This facilitated the precise exploration of the effects of diverse fertilization treatments on corn growth and soil characteristics.

### 2.2. Experimental Design

The experiment was carried out in the greenhouse of the Chinese Academy of Sciences, Huai Zhong Road, Shijiazhuang City, Hebei Province (38.02863° N, 114.53505° E), from October 2022 to January 2023. The temperature inside the greenhouse was controlled between 25 and 35 °C and the humidity was maintained at 60–70%. There was natural light and LED supplementary lighting that automatically turned on when the light intensity was below 5000 lux. The greenhouse was also equipped with environmental conditions such as ventilation, a carbon dioxide supply, and an irrigation system. The experimental soil, upon undergoing the removal of impurities such as deceased leaves and roots, was passed through a 2-millimeter sieve and thereafter placed into plastic pots with dimensions of 260 mm × 200 mm, attaining a soil mass of 8 kg per pot. In this pot experiment, the application amounts of fertilizers were measured in units per area. Five treatments were set up in the experiment: no fertilization (CK), conventional chemical fertilizer (HF1, applying 100% chemical fertilizer with a total nitrogen amount of 300 kg·hm^−2^), biogas slurry replacing 30% of the chemical fertilizer nitrogen (HF2, applying 70% chemical fertilizer with a total nitrogen amount of 300 kg·hm^−2^), biogas slurry replacing 60% of the chemical fertilizer nitrogen (HF3, applying 40% chemical fertilizer with a total nitrogen amount of 300 kg·hm^−2^), and biogas slurry replacing 100% of the chemical fertilizer nitrogen (HF5, with a total nitrogen amount of 300 kg·hm^−2^). Each treatment had four replicates.

### 2.3. Soil Sample Collection and Processing

On the 30th, 60th, 90th, and 120th days after sowing, the surface soil of potted plants under various experimental treatments was collected. In each pot, sampling was conducted according to the quartering method. After removing the litter layer, a soil sampler was used to collect soil samples from the 0–20 cm soil layer. The extracted soil was thoroughly mixed and impurities such as plant roots and stones were removed. Subsequently, a portion of the soil samples was collected in sterilized EP tubes according to the experimental treatment and stored in a constant temperature refrigerator at −20 °C for subsequent microbial determination. Another portion of the soil samples was packed in self-sealing bags according to the experimental treatment for the determination of soil nutrients in the later stage.

### 2.4. Test Methods for Physicochemical Properties

Determination of Soil Physicochemical Properties: The pH was determined by the electrode method. The determination of soil organic matter (SOM) was carried out by means of the potassium dichromate oxidation–ferrous sulfate titration approach. The total nitrogen (TN) was gauged via the semi-micro Kjeldahl method. The ammonium nitrogen (NH_4_⁺) and nitrogen in the form of nitrate (NO_3_^−^) were extracted with 1 mol·L^−1^ KCl and subsequently analyzed by a flow injection analyzer. The available phosphorus (AP) was extracted using 0.5 mol·L^−1^ NaHCO_3_ and determined by the molybdenum–antimony anti-colorimetric technique. The available potassium (AK) was extracted with 1 mol·L^−1^ NH_4_OAc and measured through the flame photometry method [13].

Determination Methods for Physicochemical Properties of Biogas Slurry: The pH was determined by the electrode method. For the determination of total nitrogen (TN), the Kjeldahl digestion (TK) method was employed. The total phosphorus (TP) was measured by the molybdenum–antimony anti-colorimetric method. The total potassium was determined through the atomic absorption method. The heavy metals were analyzed using the atomic absorption spectrophotometry method [14].

### 2.5. Extraction, Amplification, and Quantification of Soil Microbial Genomic DNA

In accordance with the kit instructions, the OMEGA M5636-02 soil DNA kit (OMEGA Bio-Tek, Norcross, GA, USA) was used to extract the total genomic DNA of soil samples. The quality and quantity of the extracted DNA samples were detected by means of a Nanodrop NC2000 spectrophotometer (Thermo Scientific, Waltham, MA, USA), a gel agarose electrophoresis instrument (DYY-6C, Beijing, China), and a gel imaging system (BG-gdsAUTO130, Beijing, China) and then they were stored at −20 °C for future use. For the V3–V4 (a) region of the bacterial 16S rRNA gene, the forward primer 338F (5′-ACTCCTACGGGAGGCAGCA-3′) and reverse primer 806R (5′-GGACTACHVGGGTWTCTAAT-3′) were used for PCR amplification. For the ITS1 (a) region of fungi, the forward primer ITS5 (5′-GGAAGTAAAAGTCGTAACAAGG-3′) and reverse primer ITS2 (5′-GCTGCGTTCTTCATCGATGC-3′) were used, and a sample-specific 7-bp barcode was added to the primers for subsequent multiple sequencing. A PCR instrument (ABI 2720, USA) was used to amplify the target gene fragment. The PCR reaction system (25 μL) contained 5 μL reaction buffer (5×), 5 μL GC buffer (5×), 0.25 μL Q5 DNA polymerase (5 U/μL), 2 μL dNTPs (2.5 mM), 1 μL forward primer (10 μM), 1 μL reverse primer (10 μM), 2 μL DNA template, and 8.75 μL ddH₂O. The thermal cycling program was as follows: initial denaturation at 98 °C for 2 min, followed by denaturation at 98 °C for 15 s, annealing at 53 °C for 30 s, extension at 72 °C for 30 s for a total of 25–30 cycles, extension at 72 °C for 5 min, and, finally, incubation at 10 °C. Vazyme vhtstm DNA clean beads (Vazyme, Nanjing, China) were used to purify the PCR amplification products. The Quant-iTPicoGreen dsDNA detection kit (Invitrogen, Carlsbad, CA, USA) was used to quantify the purified amplicons to check whether the amplification results met the sequencing requirements. After quantifying individual samples, all sample amplicons were diluted to the same concentration of 20 ng/μL and then the DNA samples were sent to a sequencing company for high-throughput sequencing. For the microbial determination of soil samples, high-throughput sequencing was used for the on-machine library construction and sequencing of collected soil samples. The determination indicators included the 16S rRNA bacterial sequencing and fungal ITS sequencing of soil samples. QIIME2 (2019.4) was used to conduct further microbial bioinformatics analysis on the sequenced data. For specific operations, refer to the official tutorial (http://docs.qiime2.org/2019.4/tutorials/, accessed on 10 September 2023). First, the primer fragments of the sequences were removed and the sequences that did not pair with the primers were eliminated. Then, the Dada2 method was used for quality filtering, denoising, splicing, chimera removal, singleton removal, and other processes to obtain the ASV characteristic sequences of fungi and bacteria in the samples. The Silva_132 database (Release 132, http://www.arb-silva.de, accessed on 10 September 2023) was used as a reference for the taxonomic annotation of bacterial characteristic sequences in the samples and the Unite_8.0 database (Release 8.0) was used as a reference for the taxonomic annotation of fungal characteristic sequences in the samples. The rarefaction depth was set to 95% of the minimum sample sequence quantity, and the ASV characteristic sequences of the samples were rarefied to meet the needs of subsequent analysis. Taxonomic composition analysis was carried out at different taxonomic levels and Alpha diversity and Beta diversity analysis plugins were used to analyze the species richness, biodiversity within each sample group, and biodiversity between groups. Differential marker species and differentially contributing species were screened, and the distribution differences of differential species among samples were further compared by means of species composition heat maps.

### 2.6. Data Processing and Analysis

Excel 2016 and IBM SPSS statistic22.0 were used for data processing and analysis. GraphPad Prism8, QIIME2 (2019.4), and R language 4.4.1 were used for results analysis and visualization. LSD minimum significant difference analysis and Duncan’s test in one-way ANOVA variance analysis were used for multiple comparisons of significant differences between treatments (*p* < 0.05). R package Corr-Version 0.43 was used for correlation analysis.

## 3. Results and Analysis

### 3.1. Growth and Nutrient Response Patterns of Silage Corn

Silage corn serves as a high-quality feed source in the livestock industry. In this study, plant height was employed to represent yield. In the pot experiment, significant differences in plant height among different treatment groups were observed 30 days after sowing (*p* < 0.05), with the biogas slurry replacing chemical fertilizer treatment exhibiting an equivalent or even superior effect. At 60 days, the plant height of corn demonstrated a trend of initially increasing and then decreasing with the escalation of the biogas slurry application rate, and the plant height in the HF2 treatment reached its peak. At this juncture, there was no significant disparity in plant height among the HF2, HF3, and HF4 treatments (*p* > 0.05). By 90 days after sowing, the plant height for the combined application of biogas slurry and chemical fertilizer was markedly higher than that for the treatment with only chemical fertilizer. The HF2 treatment consistently maintained the highest plant height throughout all stages of the experiment. Moreover, with the augmentation of the proportion of biogas slurry substituting the chemical fertilizer, the plant height (Figure 1) exhibited a decreasing trend of HF2 > HF3 > HF4, signifying the existence of a relatively appropriate ratio for the combined application of biogas slurry and chemical fertilizer to foster the growth of corn plant height.

The results of nutrient determination(Figure 1) revealed that different substitution ratios of biogas slurry exerted significant influences on the nitrogen, phosphorus, and potassium contents in stems and leaves (*p* < 0.05). The nitrogen content accumulated during the early growth stage of corn and peaked at 60 days. At this moment, the total nitrogen contents in the HF2 and HF3 treatments were the highest, followed by the HF1 treatment and the HF4 treatment. At the harvest stage (120 days), the higher the application amount of biogas slurry, the higher the nitrogen content, with the HF4 treatment increasing the total nitrogen content by 38.79% compared to the HF1 treatment. Regarding the phosphorus content, the HF2 and HF3 treatments were relatively higher at 30 days, and at the harvest stage, the phosphorus content in the HF2 treatment was significantly higher than that in other treatments. For the potassium content, it accumulated after emergence and reached its peak at 60 days. The total potassium contents in the HF1, HF2, and HF3 treatments were the highest at this stage, and at the harvest stage, the HF4 treatment increased the total potassium content by 78.39% compared to the HF1 treatment.

In Figure 2, the changes in soil nutrient content in different periods are presented. Different proportions of biogas slurry substituting for chemical fertilizer had a significant impact on soil electrical conductivity (*p* < 0.05). For example, in the harvest period (120 days), the soil electrical conductivity of the HF2 treatment reached 421.5 μs·cm^−1^, which was 87.33% higher than that of CK (225 μs·cm^−1^). As corn grew, the soil electrical conductivity gradually decreased, and the differences among treatments gradually reduced. During the growth period of corn, the content of soil organic matter decreased. The treatments with biogas slurry application were higher than those with pure chemical fertilizer treatment. At the harvest period, each fertilization treatment increased by 7.89–20.53% compared with CK. HF3 and HF4 had better improvement effects. The content of soil total nitrogen was affected by the combined application of biogas slurry. In the harvest period, HF4 > HF1 > CK, and the highest content of HF2 was 1.81 g·kg^−1^. The content of soil ammonium nitrogen decreased as corn grew. At 30 days, HF2 was 2.20 mg·kg^−1^, which was significantly increased compared with CK (0.29 mg·kg^−1^). In the harvest period, HF2 was still the highest. The content of nitrogen in the form of nitrate in soil was significantly affected by the substitution amount of biogas slurry at different growth stages. The content of soil available phosphorus decreased with the growth of crops. During the experimental stage, it first increased and then decreased with the application amount of biogas slurry. In the harvest period, HF2 was 34.39 mg·kg^−1^, which was 58.65% higher than that of CK. The phosphorus supply capacity of combined applications of biogas slurry was stronger than that of chemical fertilizer treatment. The content of soil available potassium decreased as crops grew and increased with the increase in biogas slurry application amount. At 30 days, pure application of biogas slurry (HF4) was 417.33 mg·kg^−1^, which was 122.10% higher than that of CK. In the harvest period, pure application of biogas slurry (HF4) was 258.70 mg·kg^−1^, which was 58.18% higher than that of CK. This indicated that biogas slurry contributed more to available potassium than the chemical fertilizers.

### 3.2. Characteristics of Changes in Soil Microbial Community Structure in the Harvest Period of Silage Corn

#### 3.2.1. Analysis of Soil Microbial Diversity

To deeply explore the impact of different fertilization strategies on the diversity of soil microbial communities in the harvest period, Alpha diversity analysis was conducted on soil bacteria and fungi of each treatment at 120 days. The Chao1 index was used to measure species richness, the Shannon index was used to characterize species diversity, and the coverage index reflected community coverage. The Duncan test method was used to test the significance of differences between treatments. The results showed that in terms of bacteria (see Table 2), the coverage index of all treatments reached 96%, which meant that the sequencing results represented the real situation of microorganisms. For the Chao1 index, the bacterial species richness of the HF2 and HF3 treatments was significantly higher than that of other treatments. Among them, HF2 was 6.44% higher than CK and 1.97% higher than HF1. The species richness decreased with the increase in biogas slurry application amount. For the Shannon index, the bacterial species diversity of the HF3 and HF4 treatments was significantly higher than that of other treatments. HF4 was 1.84% higher than CK and 0.76% higher than HF1. Species diversity increased with the increase in biogas slurry application amount. The treatment group with biogas slurry application increased the bacterial species diversity compared with the treatment with pure chemical fertilizer application. In terms of fungi (see Table 3), the coverage index of all treatments reached 99% and the sequencing results were reliable. For the Chao1 index, the fungal species richness of the HF2 treatment was higher than that of other treatments, being 8.99% higher than CK and slightly higher than HF1, but not significantly. The species richness decreased with the increase in biogas slurry application amount. The Shannon index showed that the fungal species diversity of the HF2 treatment was higher than that of other treatments, being 9.0% higher than CK. The HF4 treatment was lower than all other treatments. The treatment group with biogas slurry application increased the fungal species diversity compared with the treatment with pure chemical fertilizer application, but excessive biogas slurry led to a decrease in fungal species diversity.

Beta diversity can characterize the species diversity and differences between habitats. To explore the differences in the community structure and diversity of soil bacteria and fungi at 120 days under different fertilization strategies among the samples, ANOSIM was used for inter-group difference tests based on Bray–Curtis distance and NMDS analysis was carried out. For the NMDS analysis of soil bacterial OTUs at 120 days (Figure 3, left), the order was sorted according to the Bray–Curtis distance between different treatments to reflect the sequential relationship between treatments. Among them, CK and HF1 were close in distance and HF2 and HF3 were close in distance, indicating that, at this time, the bacterial communities of CK and HF1 were similar and the bacterial communities of HF2 and HF3 were similar. However, the bacterial communities of CK, HF1, and HF4 were quite different. The stress value of this analysis was 0.097 (it is advisable for it to be less than 0.2), indicating that the analysis was reasonable. The stress value of the NMDS analysis of fungi at 120 days (Figure 3, right) was 0.09, showing that the sample had practical explanatory significance and representativeness for fungi and that the model was reasonable. Moreover, the sample points within the same group of each treatment were enriched, the sample repeatability was strong, and the samples of different treatments were dispersed, indicating that different fertilization treatments had a significant impact on the changes in fungal community structure.

#### 3.2.2. Characteristics of Changes in Soil Microbial Community Structure

To deeply analyze the differential impacts of different fertilization strategies on the soil microbial community structure of potted corn, species composition analyses of soil bacteria and fungi for each treatment were conducted at the phylum level and genus level, and bar charts are used for display (Figure 4 and Figure 5). In terms of bacteria, the dominant phyla included Proteobacteria, Actinobacteria, etc. The sum of the relative abundances of the top five bacterial phyla accounted for between 73.39% and 76.64% of the total bacterial community of each treatment and the top ten bacterial phyla accounted for 95.76% to 96.72%. The application of biogas slurry led to a downward trend in the abundance of Proteobacteria. For example, HF3 decreased by 19.50% and 18.39%, respectively, compared with CK and HF1, while the abundance of Actinobacteria increased. HF4 increased by 77.38% and 27.56%, respectively, compared with CK and HF1. At the genus level, the relative abundances of the top ten bacterial genera accounted for 21.77% to 29.45% of the total bacterial community for each treatment. There were significant differences between Arthrobacter and Bacillus among treatments. HF2 and HF3 significantly reduced the abundance of Arthrobacter by 46.39% and 52.61% compared with HF1, and HF4 significantly increased the abundance of Bacillus by 139.27% compared with HF1.

In terms of fungi, the dominant phyla were Ascomycota, Basidiomycota, etc. The sum of the relative abundances of the top five fungal phyla accounted for 97.46% to 98.92% of the total fungal community for each treatment. The relative abundance of Ascomycota was the highest, ranging from 77.17% to 83.01%. The application of biogas slurry treatment increased the relative abundance of Ascomycota compared with pure chemical fertilizer application. HF4 increased by 7.25% compared with HF1. Each fertilization treatment increased the relative abundance of Mortierellomycota compared with CK, and it first increased and then decreased with the increase in biogas slurry application amount, reaching a peak at HF2. The sum of the relative abundances of the top five dominant fungal genera accounted for 39.56% to 53.66% of the total fungal community for each treatment. The relative abundance of Gibberella was the highest. The application of biogas slurry treatment increased the relative abundance of Gibberella compared with pure chemical fertilizer application. HF4 increased by 84.49% compared with HF1. With the increase in biogas slurry application amount, the relative abundance of Chaetomium decreased. HF4 was 56.02% lower than HF1.

#### 3.2.3. Key Populations with Differences in Microbial Communities

By means of LEfSe analysis, LDA > 3 and *p* < 0.05 were set as the differential screening threshold to obtain species with significant differences in relative abundance between groups. The relevant situation is shown in Figure 6 (above). In terms of bacteria, a total of 54 significantly different populations were identified. Among them, there were 5 in the CK treatment, 5 in the HF1 treatment, 8 in the HF2 treatment, 7 in the HF3 treatment, and 29 in the HF4 treatment. Regarding the main differential species (Figure 6, below), the main differential species in the CK treatment included *Vicinamibacterales*, *Xanthobacteraceae*, *Hyphomicrobiaceae*, *and Pedomicrobium*; the main differential species in the HF1 treatment were *Gaiellales* and *Cyanobacteriia*; the main differential species in the HF2 treatment were *KD4-96*, *Clostridia*, *Luteimonas*, *Planococcaceae*, and *Devosia*; the main differential species in the HF3 treatment were *Gemmatimonadota*, *Ellin6067*, *Azoarcus,* and *Rhodocyclaceae*; the main differential species in the HF4 treatment included *Bacteroidota*, *Bacteroidia*, *Micrococcales*, *Arthrobacter*, *Sphingomonadaceae*, *Cytophagales*, *Nocardioidaceae*, *Solirubrobacteraceae*, *Anaerolineae*, *Blastococcus*, *Polyangia*, *Blastocatellaceae*, *Altererythrobacter*, *Peptostreptococcales-Tissierellales*, *Verrucomicrobiota*, *Peptostreptococcaceae*, *Novosphingobium*, *Nitrosospira*, and *Romboutsia*. In addition, when LDA > 4 and *p* < 0.05 were used as the differential screening threshold in LEfSe analysis, species with significant differences in relative abundance between groups were also obtained, as shown in Figure 7 (left). In terms of fungi, a total of 19 significantly different populations were identified. Among them, there were two in the CK treatment, one in the HF1 treatment, nine in the HF2 treatment, three in the HF3 treatment, and four in the HF4 treatment. For the main differential species (Figure 7, right), the main differential species in the CK treatment were *Eurotiales* and *Penicillium*; the main differential species in the HF1 treatment was *Onygenales*; the main differential species in the HF2 treatment were *Sordariales*, *Chaetomiaceae*, *Chaetomium*, and *Schizothecium*; the main differential species in the HF3 treatment were Pithoascus, Microascaceae, and *Microascales*; the main differential species in the HF4 treatment included *Gibberella*, *Nectriaceae*, *Hypocreales*, and *Sordariomycetes*.

### 3.3. Correlation Between Microbial Community Diversity and Soil Physicochemical Properties

Pearson correlation analysis was used to study the relationship between soil bacterial community diversity and soil physicochemical properties. As shown in Table 4, the Chao1 index of bacterial and fungal communities was significantly positively correlated with AP, AK, and NO_3_^−^-N (*p* < 0.05) and significantly negatively correlated with pH (*p* < 0.05). The Shannon index of bacteria was extremely significantly positively correlated with EC, AP, AK, TN, and NO_3_^−^-N (*p* < 0.001) and extremely significantly negatively correlated with pH. The Shannon index of fungi was extremely significantly positively correlated with SOM and TN.

## 4. Discussion

### 4.1. Impact of Combined Application of Biogas Slurry on Silage Corn and Soil Physicochemical Properties

In the experiment, the corn’s plant height rose swiftly initially and then plateaued. At 40% biogas slurry substitution (HF2), the growth promotion was most notable. But, as the substitution ratio climbed further, the plant height declined. Evidently, sufficient biogas slurry supplies corn’s N, P, and K needs, facilitating growth. Complete substitution, however, may cause nutrient imbalance, impeding growth, corroborating Wang Wei’s findings [15].

N, P, and K are crucial nutrients for plants, and the combined application of biogas slurry and chemical fertilizers exhibits unique advantages. In rigorous experimental comparisons, the substitution of chemical fertilizers with biogas slurry significantly outperformed the single application of either chemical fertilizer or organic fertilizer, substantially enhancing the absorption amounts of nitrogen, phosphorus, and potassium in corn plants. Moreover, the optimal range of the substitution ratio of organic fertilizers was precisely determined. The cutting-edge research by Liu Yidan and Xiuli Xin [16,17] provides strong corroboration. In this study, when 30–70% of chemical fertilizers were replaced by biogas slurry, it effectively augmented fertilizer effectiveness; vigorously curtailed the losses of nitrogen, phosphorus, and potassium; deeply stimulated the nutrient absorption vitality of crops; and drove the accumulation of nitrogen, phosphorus, and potassium in plants to ascend. This aligns seamlessly with the findings of Wu Chenglong’s [18] 15N labeling technique in detecting nitrogen accumulation in winter wheat, Ren Keyu’s exploration of reducing phosphorus losses and increasing phosphorus content through 50% substitution, and Zhou Jiangming’s [19] revelation of enhancing potassium accumulation and grain transfer for yield increase in crops. It robustly verifies that biogas slurry plays a pivotal role in empowering nutrient absorption in corn.

Electrical conductivity, as a crucial indicator representing the content of soluble salts in the soil, can reflect the electrochemical and ion separation characteristics of the soil. It is based on the ion migration in the soil pore solution, and the faster the ion migration, the higher the electrical conductivity [20]. This study demonstrated that in the early stage of the experiment, each fertilization treatment significantly increased the soil electrical conductivity. In the final stage, the treatment with only chemical fertilizers (HF1) decreased to the level of the control group, while the treatments with biogas slurry application (HF2, HF3, HF4) remained significantly higher than the control group, mainly because biogas slurry contains a large amount of salt-based ions such as sodium and chloride ions. The long-term or excessive application of biogas slurry can cause soil salt accumulation, resulting in salt damage, inhibiting crop growth, and even leading to crop death in severe cases [21,22].

Soil organic matter content is a key indicator for measuring soil fertility, which is of great significance for stabilizing soil structure and stimulating soil biological activity and is one of the core elements for maintaining soil health and promoting crop yield increase [23]. In the early stage of the experiment, the soil organic matter content increased with the increase in the amount of biogas slurry application; in the later stage, there was no significant difference among the treatment groups with biogas slurry application (HF2, HF3, HF4), but they were significantly higher than the treatment with only chemical fertilizers (HF1). This was due to the fact that substances such as cellulose in biogas slurry were converted into organic matter under the action of microorganisms and supplemented into the soil. This is in line with the research results of Wang Jinhua et al. [24], indicating that the application of organic substances can not only promote the decomposition of organic matter but also increase its content, improve the quality of humus, and further enhance soil’s nutrient supply capacity [25].

Soil nitrogen inventory and nitrogen form distribution are related to a soil’s nutrient supply and retention capacity; the combined application of biogas slurry and chemical fertilizer had an improving effect on soil nitrogen fertility. In the pot experiment, the soil total nitrogen content in the control and the treatment with only chemical fertilizers gradually decreased, while the treatment groups with biogas slurry application maintained a relatively high level in each period. Li Shushan et al. [26] found that there was a significant positive correlation between the application of chemical fertilizer nitrogen and the increase in soil nitrogen content in the form of nitrate. In the treatment with only chemical fertilizers, 27% of the nitrogen in the form of nitrate originated from the transformation of exogenous chemical fertilizer nitrogen. In contrast, in the treatment with combined application of organic fertilizers, only 5% originated from the transformation of exogenous organic fertilizer nitrogen. Some studies have demonstrated that during the wheat harvest season, the content of nitrogen in the form of nitrate in the soil decreases as the proportion of biogas slurry nitrogen substitution increases. This might be due to the fact that organic fertilizers can improve the content of soil aggregates and active carbon, thus enhancing the retention ability of nitrogen in the form of nitrate. Moreover, the increase in the soil carbon-to-nitrogen ratio after the application of organic fertilizers stimulates microbial activity to retain inorganic nitrogen. The results of this experiment are similar to those reported in the relevant literature [27]. The ammonium nitrogen content in the HF1 treatment showed a cliff-like decrease on the 30th and 60th days, which may be because urea is rapidly converted into ammonium nitrogen in the short term after being applied to the soil, and, in the early stage of fertilization, biogas slurry contains abundant ammonium nitrogen, which is beneficial for rapid absorption and utilization by corn, resulting in a relatively high content 30 days after emergence and a significant decrease in the middle stage. In the later stage, the decomposition of biogas slurry organic fertilizer is slow, and the nitrogen supply capacity is persistent. In the later stage of the pot experiment, the soil ammonium nitrogen content in the treatment with biogas slurry application was significantly higher than that in the treatment with only chemical fertilizers.

Numerous studies have shown that the application of biogas slurry in agricultural production can significantly increase the content of phosphorus and potassium elements in the soil to meet the nutrient requirements of crops [28,29]. In each period of the pot experiment, the available phosphorus content showed a decreasing trend with the increase in the amount of biogas slurry application, which may have been because the biogas slurry was in a liquid state and the retention amount of the surface soil was limited. Excessive application could cause runoff loss. That is, the application of biogas slurry had a negative effect on the soil’s available phosphorus content, which was similar to the research result of Wang Weiping [30]. In the pot experiment, the potassium element mainly came from biogas slurry and chemical fertilizer, and the potassium content in the biogas slurry was much higher than that in the chemical fertilizers. With the increase in the amount of biogas slurry application, the soil available potassium content gradually increased, which is similar to the research conclusion of Xiao Yang [31]; this could have been caused by the capillary salt extraction effect due to soil water evaporation. The potassium element can promote the synthesis of nucleic acids and proteins, enhance cell water retention capacity, improve water use efficiency, and enhance the drought resistance of corn.

### 4.2. Succession Characteristics of Soil Microbial Community Structure of Silage Corn Under the Combined Application of Biogas Slurry

Generally, the application of biogas slurry helps to improve the soil microbial functional diversity index [32,33], which is consistent with the conclusion of this study. Compared with the control (CK), the fertilization treatment increased the soil microbial richness and diversity; compared with the single application of chemical fertilizers, an appropriate amount of biogas slurry further improved the richness and diversity of soil bacteria and fungi, but when biogas slurry completely replaced chemical fertilizers, the richness and diversity of both decreased. In various agricultural ecosystems, there are reports indicating that biogas slurry has a promoting effect on soil bacterial diversity [34]. Studies have pointed out that the exogenous carbon in biogas slurry can enhance the activity of soil microorganisms in the short term, thereby causing changes in their quantity and community structure [35]. The NMDS analysis based on the Bray–Curtis algorithm showed that different fertilization strategies under the same nitrogen amount have a significant impact on the soil bacterial and fungal community structures. The application of biogas slurry can improve the soil microenvironment to a certain extent, which may be beneficial to the growth and reproduction of some bacterial and fungal groups, resulting in differences in fungal diversity in individual scale or quantity, and finally leading to changes in community structure. The results of the analysis of bacterial and fungal species composition in this study indicated that there were varying degrees of changes in the soil microbial community composition among the different treatments.

From the perspective of the community composition of dominant species at the phylum level, although different fertilization strategies changed their distribution proportions to a certain extent, they had little impact on the composition of dominant bacterial and fungal phyla in the soil. At the bacterial phylum level, the top five dominant phyla in each treatment were Actinobacteria, Proteobacteria, Acidobacteria, Firmicutes, and Chloroflexi, which were similar to the research results of Xing Pengfei and Li et al. [36,37]. At the fungal phylum level, the top four dominant phyla in each treatment were Ascomycota, Basidiomycota, Mortierellomycota, and Mucoromycota. The differences among the treatments indicated that the bacterial and fungal community compositions in the treatments with chemical fertilizers and no fertilization were significantly different from those in the treatments with biogas slurry application. Compared with the single application of chemical fertilizers, after the application of biogas slurry, the relative abundances of Actinobacteria and Ascomycota increased, while the relative abundance of Proteobacteria decreased. Previous studies have shown that Actinobacteria can decompose organic matter, accelerate the carbon cycle, promote the formation of soil aggregates, and secrete plant growth hormones [38]; Proteobacteria contains nitrogen-fixing bacteria and photosynthetic bacteria related to plant growth and soil fertility, which can promote nitrogen fixation in soil and plants and activate the fixed phosphorus in soil to increase the available phosphorus content [39]; Ascomycota mainly decomposes organic matter and degrades soil residues, promotes soil nutrient cycling, but contains pathogenic bacteria [40]. In many studies with different soils and crops as research objects, Actinobacteria, Proteobacteria, Acidobacteria, Ascomycota, and Basidiomycota dominated in the soil habitat, and the results of this study are similar. In the soil samples of all treatments, these bacterial and fungal phyla as dominant species occupied most of the soil microbial niches.

From the perspective of the community composition of dominant species at the genus level, there were varying degrees of changes in the species composition and abundance proportion of soil bacteria and fungi. Compared with the treatment with only chemical fertilizers, the treatment with biogas slurry application increased the relative abundances of Bacillus and Gibberella in the soil and decreased the relative abundances of Arthrobacter and Chaetomium. According to previous studies, Bacillus can promote the decomposition of organic matter, drive the soil material cycle and nutrient transformation, antagonize pathogenic bacteria, reduce soil-borne diseases, and promote crop growth [41]; Gibberella can promote the growth and development of crops, make them mature earlier, increase yield and improve quality, and quickly break the dormancy of organs such as seeds and promote germination; Arthrobacter can produce nitrite reductase to reduce the harm of nitrite; Chaetomium can survive and reproduce under environmental pressures such as low temperatures and low oxygen levels.

This study conducted a correlation analysis between the genus level of the main soil bacterial community and soil physicochemical properties. The results showed that there is a relationship between the relative abundances of different soil microbial bacterial genera and soil nutrient contents. Specifically, genera were extremely significantly positively correlated with EC, AP, AK, TN, and nitrogen in the form of nitrate (*p* < 0.001) and extremely significantly negatively correlated with pH. The fungal Shannon index had an extremely significant positive correlation with SOM and TN. Among these indicators, pH was significantly different from the other indicators. It is speculated that this was due to the buffering capacity of the soil. Because the soil contained abundant cations and anions and had a large exchange capacity, when biogas slurry was applied, it may have reacted with some substances through exchange, precipitation, etc. [42]. Compared with the soil acidification caused by the long-term excessive application of chemical fertilizers, the combined application of biogas slurry in a reasonable proportion has no significant impact on soil pH, and its specific mechanism still needs further exploration.

## 5. Conclusions

Research reveals that substituting 30% chemical fertilizer with biogas slurry presents an optimal comprehensive effect, enhancing corn nutrient uptake, soil fertility, and microbial health. Compared to sole chemical fertilization, the combination of biogas slurry and fertilizers boosts SOM, AP, and AK contents. Biogas slurry enriches soil microbial diversity; the bacterial Alpha diversity peaks at 30% substitution. Beta diversity analysis shows significant differences in bacterial community structures among fertilization strategies, reflecting distinct responses to soil changes. Correlation analyses showed that microbial diversity indices related to soil nutrients and properties. These findings underpin biogas slurry utilization, fertilizer reduction, nutrient loss, and soil degradation mitigation in the agro-pastoral transition zone, propelling agricultural green progress.

## Figures and Tables

**Figure 1 microorganisms-13-00002-f001:**
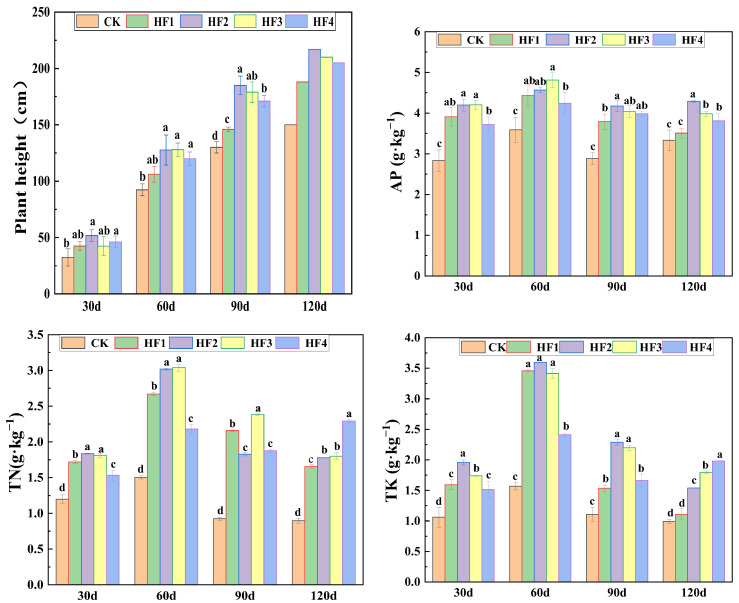
Plant height of plants at different times with different substitutions of biogas slurry for chemical fertilizers. Different lowercase letters indicate significant differences among different treatments (*p* < 0.05).

**Figure 2 microorganisms-13-00002-f002:**
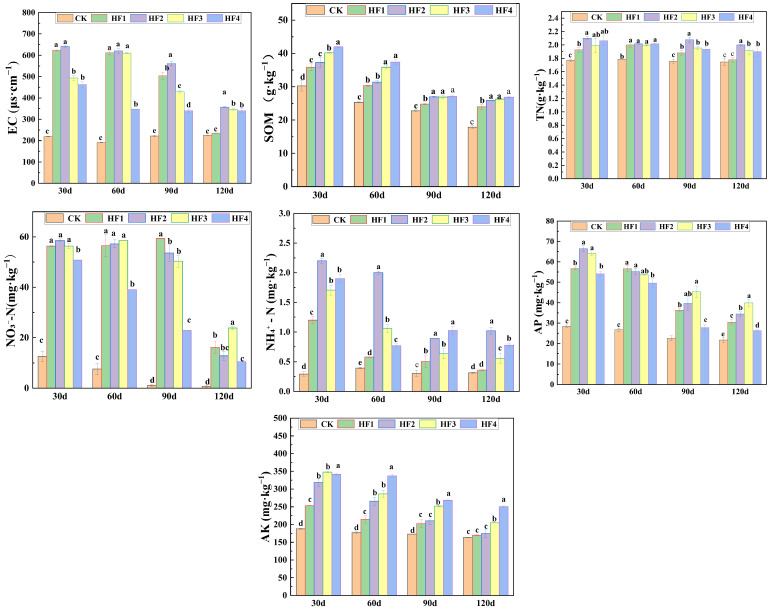
Variations in soil nutrient content in different periods when different amounts of biogas slurry replaced chemical fertilizers. Different lowercase letters indicate significant differences among different treatments (*p* < 0.05).

**Figure 3 microorganisms-13-00002-f003:**
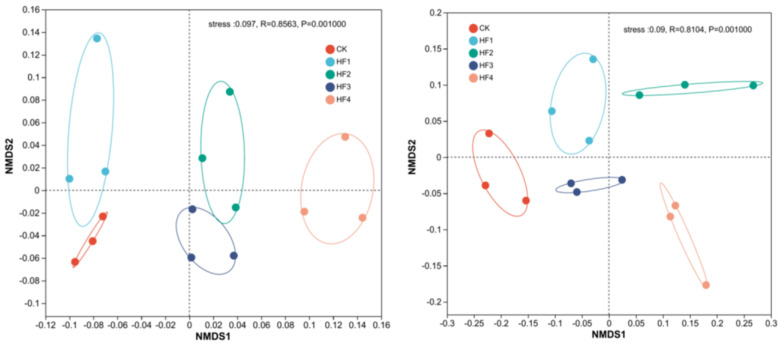
Non-metric multidimensional scaling (NMDS) analysis of soil bacterial (**left**)/fungal (**right**) community structure under different fertilization strategies based on OTU level.

**Figure 4 microorganisms-13-00002-f004:**
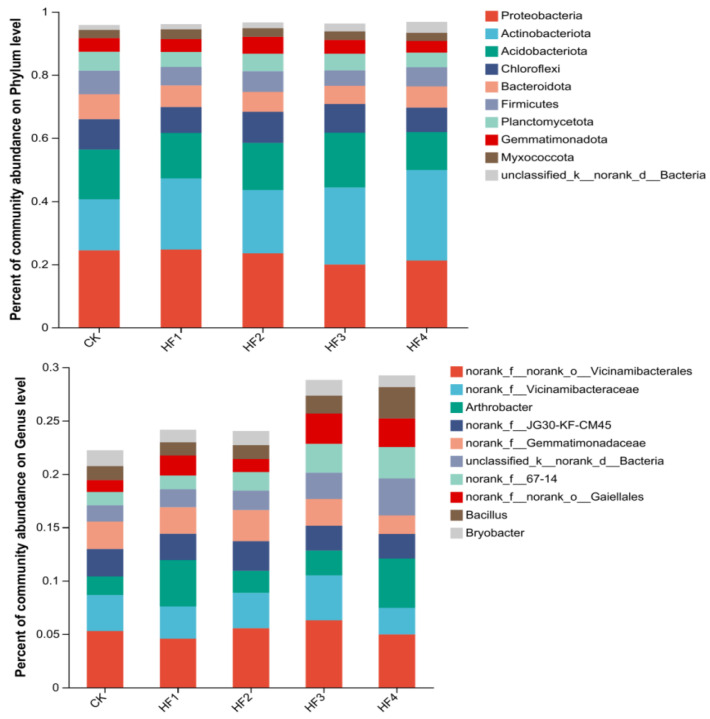
Composition of dominant bacterial phylum communities (the figure **above**) and bacterial genus communities (the figure **below**) in soil at 120 days (top 10 in relative abundance).

**Figure 5 microorganisms-13-00002-f005:**
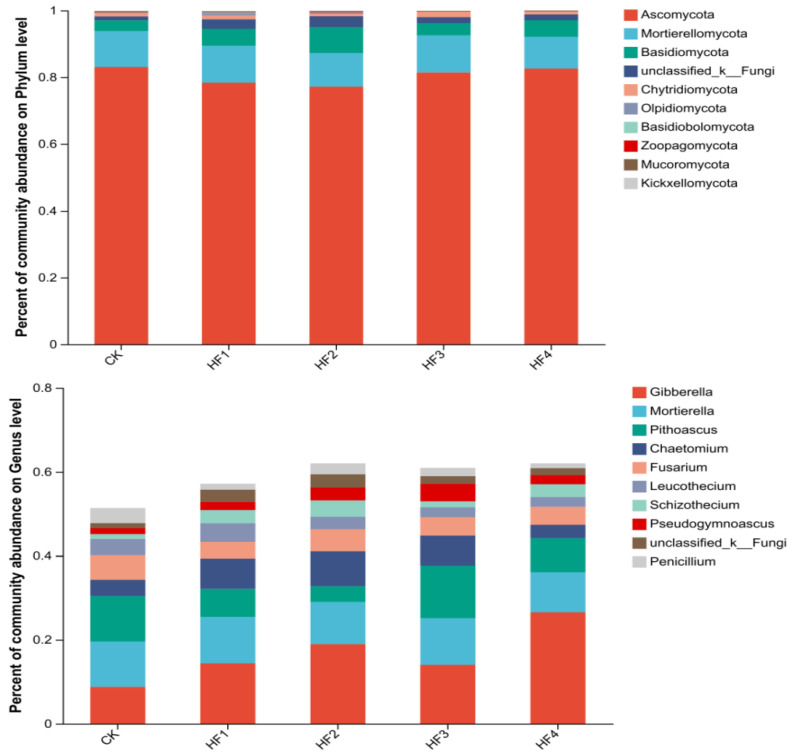
Composition of dominant fungal phylum communities (the figure **above**) and fungal genus communities (the figure **below**) in soil at 120 days.

**Figure 6 microorganisms-13-00002-f006:**
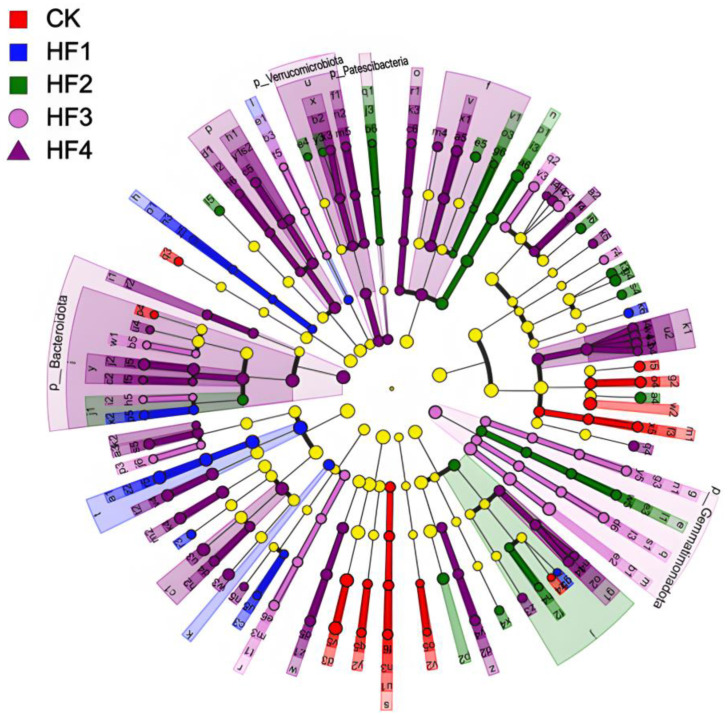
LEfSe analysis of differential soil bacterial species under different fertilization treatments. Evolutionary branch diagram (the figure **above**) and LDA value (the figure **below**).

**Figure 7 microorganisms-13-00002-f007:**
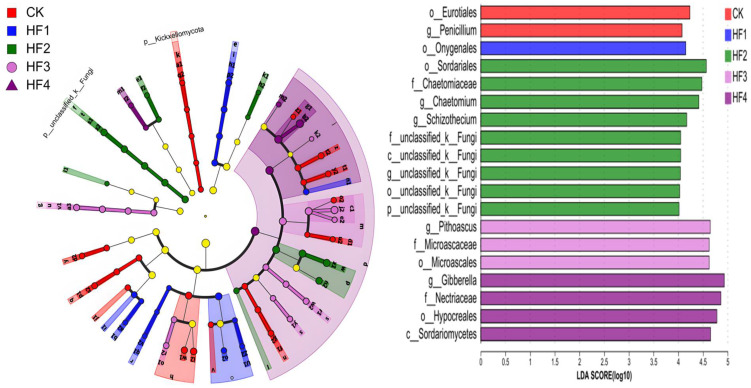
LEfSe analysis of differential soil fungal species under different fertilization treatments. Evolutionary branch diagram (**left**) and LDA value (**right**).

**Table 1 microorganisms-13-00002-t001:** Basic properties and heavy metal contents of biogas slurry.

	TN	TP	TK	pH	Ni	As	Cu
	(g·kg^−1^)	(mg·kg^−1^)	(g·kg^−1^)		(μg·L^−1^)	(μg·L^−1^)	(mg·L^−1^)
Biogas Slurry	3.38	254.00	2.54	8.20	598.09	84.78	4.98
	Zn	Cr	Cd	Pb	Fe	Ca	Mg
	(mg·L^−1^)	(μg·L^−1^)	(μg·L^−1^)	(μg·L^−1^)	(mg·L^−1^)	(mg·L^−1^)	(mg·L^−1^)
Biogas Slurry	21.47	634.50	9.28	120.96	91.48	783.98	634.76

**Table 2 microorganisms-13-00002-t002:** Alpha diversity of soil bacteria in the harvest period (120 days) of silage corn.

Treatment	Chao1	Shannon	Coverage
CK	4029.55 ± 46.01 c	6.52 ± 0.02 c	0.96 ± 0.00 a
HF1	4205.95 ± 13.98 b	6.59 ± 0.03 b	0.96 ± 0.00 a
HF2	4288.87 ± 20.64 a	6.63 ± 0.03 ab	0.96 ± 0.00 a
HF3	4266.93 ± 33.58 a	6.64 ± 0.02 a	0.96 ± 0.00 a
HF4	4151.81 ± 76.79 b	6.64 ± 0.03 a	0.96 ± 0.00 a

Note: In the same column, different lowercase letters indicate significant differences at the 5% level among treatments.

**Table 3 microorganisms-13-00002-t003:** Alpha diversity of soil fungi in the harvest period (120 days) of silage corn.

Treatment	Chao1	Shannon	Coverage
CK	556.89 ± 22.65 b	4.11 ± 0.12 b	0.99 ± 0.00 a
HF1	584.52 ± 16.40 ab	4.22 ± 0.21 ab	0.99 ± 0.00 a
HF2	606.93 ± 12.03 a	4.48 ± 0.11 a	0.99 ± 0.00 a
HF3	584.60 ± 17.07 ab	4.21 ± 0.26 ab	0.99 ± 0.00 a
HF4	557.69 ± 13.07 b	3.75 ± 0.10 c	0.99 ± 0.00 a

Note: In the same column, different lowercase letters indicate significant differences at the 5% level among treatments.

**Table 4 microorganisms-13-00002-t004:** Pearson correlation analysis of soil physicochemical properties and bacterial and fungal diversity indices.

		pH	SOM	EC	AP	AK	TN	NO_3_^−^-N	NH_4_^+^-N
bacterial	Chao1	−0.525 **	−0.100	0.375	0.685 ***	0.598 **	0.059	0.648 ***	0.191
	Shannon	−0.683 ***	0.382	0.737 ***	0.673 ***	0.752 ***	0.573 **	0.672 ***	0.32
fungal	Chao1	−0.464 *	−0.395	0.178	0.535 **	0.507 *	−0.328	0.457 *	0.246
	Shannon	−0.237	−0.615 **	−0.153	0.371	0.256	−0.669 ***	0.236	0.146

Note: *: *p* < 0.05; **: *p* < 0.01; ***: *p* < 0.001.

## Data Availability

The original contributions presented in this study are included in the article. Further inquiries can be directed to the corresponding author.

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
