# Peer review of "Effects of Combined Application of Biogas Slurry and Chemical Fertilizers on Silage Corn, Soil Nutrients, and Microorganisms"

_microorganisms, 2024, doi:10.3390/microorganisms13010002_

Round 1
Reviewer 1 Report
Comments and Suggestions for Authors
Dear Editor and Authors,
The article requires improvements as outlined below:
- Avoid using "nitrate nitrogen" and instead use "nitrogen in the form of nitrate."
- The introduction provides a solid foundation for the study; however, the manuscript does not include the study's hypothesis or objectives, which are essential in scientific work. This is particularly important as the study's conclusions are directed and aligned with its objectives.
- Please specify the methodologies used to determine both the available nutrient levels in the soil and the total content in the biogas slurry. This information is necessary to enable comparisons with other studies in the literature.
- Since the biogas slurry contains significant amounts of other elements, such as P, how did the authors account for the effect of these additional nutrients? This nutrient input directly impacts the results obtained. If this was not addressed, please include this information, calculate the input of all elements, and analyze how these correlate with the data obtained.
- In section 2.4, include the methodologies and their citations for the determinations performed.
- The quality and size of the figure lettering are too small; please improve this.
- Present the results as percentages of increase or decrease for each variable analyzed.
- Figure 6 is poorly presented, making it difficult to interpret; please revise it for clarity.
- The discussion is well-founded and supported by current literature.
- The conclusion is too long; the authors should make it more concise and ensure it aligns with the study's objectives.
Reviewer 2 Report
Comments and Suggestions for Authors
The manuscript is quite well prepared but contain some drawbacks.
Line 121: what do you mean as a “available nitrogen”. It is mineral nitrogen, NH4+ and NO3- or other forms of nitrogen?
Line 120: Please provide the type of the soil according FAO WRB classification not only as “steppe chestnut soil”.
The geographical coordinates provided in the manuscript 38.045474°N, 114.502461°E indicate the location of stadium not greenhouse. Is it correct?
The soil in the greenhouse was natural or it was it a soil substrate prepared for the experiment based on natural soil?
Line 161: Unnecessary brackets “()”?
Line 215: Why two methods of multiple comparisons were used, LSD and Duncan test? Results of what method are presented in Fig. 1 and 2 as well tables 2 and 3?
The experiment was conducted as a pot experiment. The amount of the fertilizers are provided in units per area for example for nitrogen 300 kg per hm-2. How it was performed? Could you provide some specific information how the fertilization was applied in the pot experiment?
The references are not formatted according the guidelines for authors, for example the names of the authors should not be written using capital letters.
The only trait of the plans which was observed in the study is plant height? Other traits, for example aboveground biomass of the plants, were not observed? It is important because it allow to evaluate yield potential with different fertilization as well the effect of microbial communities on the yield.
Why in the supplementary file there are the same figures and tables which are in the manuscript? It is unnecessary duplication.
Explain what the asterisks in table 4 mean.
Please use one type of language in all the manuscript, British or American only. For example in the title there is word “maize” but in the text of the manuscript there is “corn” (line 426).
Round 2
Reviewer 1 Report
Comments and Suggestions for Authors
Dear Editor and Authors,
After a careful analysis of the manuscript and the reviewers' responses, I believe the article has been significantly improved, and all my main concerns have been adequately addressed. Therefore, I have no further comments to make.